# Associations of heavy metals and urinary sodium excretion with obesity in adults: A cross-sectional study from Korean Health Examination and Nutritional Survey

**Juyeon Lee**[1,2,3], **Sue K. Park**[1,2,3]*

1 Department of Preventive Medicine, College of Medicine, Seoul National University, Jongnogu, Seoul, Republic of Korea, 2 Cancer Research Institute, Seoul National University, Jongnogu, Seoul, Republic of Korea, 3 Integrated Major in Innovative Medical Science, Seoul National University Graduate School, Jongnogu, Seoul, Republic of Korea

* suepark@snu.ac.kr

## Abstract

### Backgrounds

Sodium and heavy metals are among the key elements to which humans are exposed due to environmental pollution and food consumption. Investigating the association between heavy metals, urinary sodium excretion, and obesity is of significant importance, given potential implications for public health. Therefore, this study aims to explore the relationship between heavy metals, urinary sodium excretion, and overweight and obesity in the Korean population.

### Methods

Data from 6,609 participants in the Korea National Health and Nutrition Examination Survey (KNHANES) from 2008 to 2012 were analyzed. Associations between heavy metals (cadmium, mercury), urinary arsenic, sodium excretion levels, and overweight/obesity were assessed using polytomous logistic regression models, adjusting for confounders.

### Results

Higher mercury (OR = 1.57, 95% CI = 1.31–1.88) and urinary sodium (OR = 2.21, 95% CI = 1.84–2.66) levels are associated with an increased risk of overweight and obesity. Notably, participants with elevated levels of both mercury and urinary sodium had the highest odds of being overweight and obesity (OR = 3.61, 95% CI = 2.61–5.00). In contrast, other heavy metals, such as serum cadmium and urinary arsenic, were not statistically significantly associated with the risk of overweight and obesity.

### Conclusions

This finding highlights the intricate multiplicative effect between mercury and sodium on obesity, necessitating further research to validate and understand the connections. Heavy

website (https://knhanes.kdca.go.kr). For further assistance, please contact the KDCA at knhanes@korea.kr.

**Funding:** This study was supported by a grant from Seoul National University Hospital (2025 to S.K.P). This study was supported by the National R&D Program for Cancer Control through the National Cancer Center(NCC) funded by the Ministry of Health & Welfare, Republic of Korea (HA21C0140 to S.K.P.). The funders had no role in the design of the study, collection, analysis, interpretation of data, and writing of this article.

**Competing interests:** The authors have declared that no competing interests exist.

**Abbreviations:** KNHANES, The Korea National Health and Nutrition Examination Survey; EPA, Environmental Protection Agency; CDC, U.S. Centers for Disease Control and Prevention; IRB, Institutional review board; WHO, World Health Organization; BMI, Body mass index; WC, Waist circumference; VAT, Visceral adipose tissue.

metals, particularly mercury, exert an influence on obesity, and the possibility of an enhanced impact on obesity, especially when acting in conjunction with salt, is indicated.

## Introduction

Obesity has become a global health concern, reaching epidemic proportions in recent decades. It is characterized by excessive accumulation of body fat and is associated with numerous health complications, including cardiovascular diseases, diabetes mellitus, and musculoskeletal disorders [1,2]. The etiology of obesity is multifactorial, involving a complex interplay of genetic, environmental, and lifestyle factors [3,4].

Sodium is an essential electrolyte that regulates fluid balance and blood pressure in the body [5]. While excessive sodium intake is commonly associated with hypertension, a well-established risk factor for cardiovascular diseases [6], recent studies suggest that high dietary sodium intake may also contribute to obesity [7,8]. Some epidemiological studies have indicated that elevated sodium intake can increase appetite and food consumption, promoting weight gain [9–11].

Heavy metals are naturally occurring elements that can accumulate in the body through environmental exposures such as air pollution, contaminated water, and food sources [12]. Exposure to heavy metals like lead, cadmium, mercury, and arsenic has been linked to adverse health outcomes, including neurotoxicity, cardiovascular diseases, and endocrine disruption [13]. Emerging evidence also suggests a potential connection between heavy metal exposure and obesity [14,15]. Chronic exposure to heavy metals may disrupt metabolic processes, interfere with hormone regulation, and promote inflammation, all of which could contribute to weight gain and obesity development [16,17]. Similarly, another cross-sectional study demonstrated an association between mercury exposure and obesity [18], emphasizing the role of environmental factors in body weight regulation. Additionally, a recent meta-analysis reported consistent findings linking mercury exposure to obesity across diverse populations [14].

In this study, the rationale for combining sodium and heavy metals in the analysis is based on their strong association with obesity and their potential contribution to its development through shared pathological pathways, such as oxidative stress and inflammation [19–21]. Sodium metabolism is primarily regulated by the kidneys [5], which also play a key role in the excretion and accumulation of heavy metals [22], suggesting a potential interaction between these two factors. Most previous studies have focused on the individual effects of sodium or heavy metal exposure, with limited research examining their combined effects. Combined exposure to sodium and heavy metals is particularly relevant in populations with high-sodium diets and environmental pollution, making it a critical public health concern.

Therefore, this study aimed to investigate the relationship between urinary sodium excretion and heavy metal exposures (cadmium, mercury, and arsenic) in overweight and obesity, focusing on their individual effects. Additionally, this study aims to evaluate the combined effects of urinary sodium excretion and serum mercury levels on the risk of overweight and obesity, exploring potential interactive effects between these exposure factors.

## Material and methods

### Study populations

This study utilized data from the Korea National Health and Nutrition Examination Survey (KNHANES) conducted between 2008 and 2012 KNHANES is a nationally representative

cross-sectional survey designed to assess the health and nutritional status of the Republic of Korea population [23]. The survey employs a complex, stratified, multistage, probability sampling method to select participants from non-institutionalized civilians in South Korea. Details of KNHANES have been published elsewhere. [24]

The current cross-sectional study was restricted to participants aged ≥ 19 years who completed the health examination survey, including heavy metal measurements (n = 12,000). Among these, we excluded participants with a history of stomach cancer, liver cancer, colorectal cancer, lung cancer, stroke, or cardiovascular disease, as well as those with chronic kidney failure or chronic liver diseases at enrollment (N = 1,350). These participants were excluded due to their higher tendency to develop anorexia, which could lead to unhealthy nutrient intake patterns. Additionally, participants with chronic kidney disease were excluded to ensure the accuracy of urinary sodium and arsenic analyses. Furthermore, we excluded 4,041 participants with missing information on exposure variables (spot serum and urine) and the outcome variables, including anthropometric measurements. Therefore, the final study population consisted of 6,609 participants (Fig 1). Furthermore, for the purpose of conducting sensitivity analysis in this study, the safety limits established by the U.S. Environmental Protection Agency (EPA) and the U.S. Centers for Disease Control and Prevention (CDC) for each respective heavy metal were verified. Participants exceeding these criteria accounted for approximately 7% (472 subjects) of the total study population, and these findings are presented in the S1 and S2 Tables. This study was conducted according to the guidelines established by the Declaration of Helsinki. All participants provided written informed consent before participating in the survey. This study was a retrospective analysis of medical records (or archived samples) collected from the Korea Disease Control and Prevention Agency (KDCA). The data were accessed on [12, September, 2024]. The present study was exempted from review by the Institutional Review Board of Seoul National University Hospital (IRB No: E-2409-044-1569). The authors confirm that all data were fully anonymized before access. No author had access to information that could identify individual participants either during or after the data collection.

## Anthropometric measurements

Anthropometric measurements, including height, weight, waist circumference, and hip circumference, were obtained by trained KNHANES staff following standardized procedures [23]. Weight was measured to the nearest 0.1 kg with a GL-6000-20 (Korea G-tech Ltd.) and height was measured to the nearest 0.1 cm using a Seca 225 (Germany Seca Ltd.). Body mass index (BMI) was calculated as weight (kg) divided by height squared ($m^2$). This study categorized participants based on the World Health Organization (WHO) guidelines for Asian populations, classifying underweight as a BMI < 18.5 $kg/m^2$, normal weight as a BMI between 18.5 and 22.9 $kg/m^2$, and overweight and obesity as a BMI ≥ 23 $kg/m^2$ [25].

## Assessment of exposure: Heavy metals and calculated 24-h urinary sodium excretion

The estimation of heavy metals in the blood during the KNHANES was conducted by the Neodin Medical Research Institute. Blood samples were collected using Heparin tubes. Serum cadmium level was performed using Atomic Absorption Spectrometry method (AAnalyst 600, PerkinElmer, Finland). For the estimation of mercury in the blood, the Gold Amalgamation Technique was used, and measurements were carried out using DMA-80 (Milestone, Italy). The method detection limits were 0.081 μg/L for cadmium, and 0.05 μg/L for mercury. Urinary arsenic levels were measured with a graphite furnace atomic absorption spectrometer

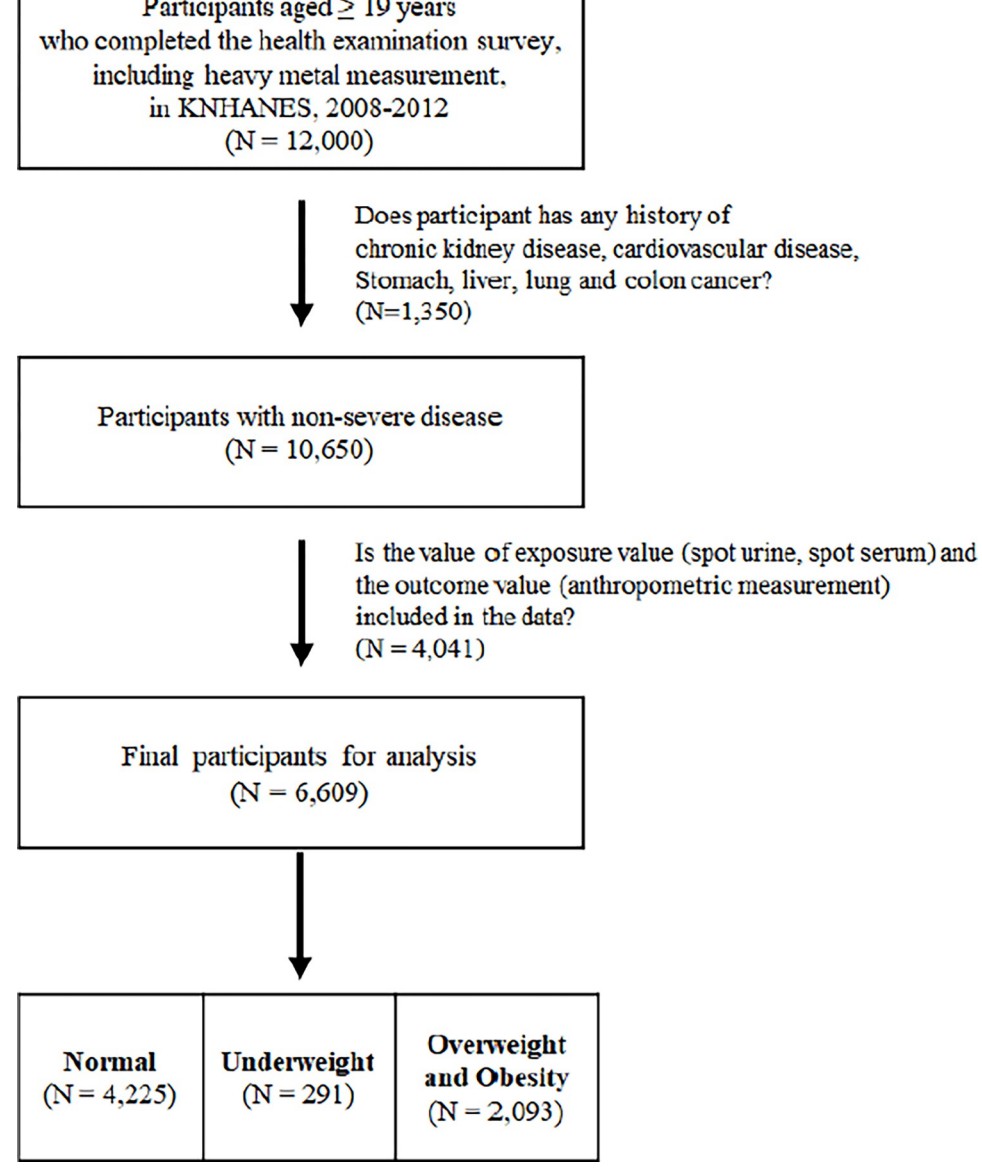

**Fig 1. Flow chart of exclusion criteria to specify the study population.**

(AAnalyst 600, PerkinElmer, Finland). The limit of detection for the arsenic level was 1.679 mg/L [23]. In this study, we investigated the biological exposure levels of heavy metals in the general population and utilized the safety limits established by CDC and EPA for serum heavy metals and urine arsenic (serum cadmium > 5 μg/L, serum mercury > 10 μg/L, and urine arsenic > 35 μg/g creatinine) (S1 and S2 Tables).

Urinary sodium and creatinine were measured in the same central laboratory for all participants. Urinary creatinine concentrations were calculated based on a colorimetric method (Hitachi Automatic Analyzer, Hitachi, Japan). Urinary sodium levels were measured using the Hitachi 7600 ISE special reagent (Japan Hitachi Ltd.) [23].

The KNHANES provided data on urine creatinine and sodium concentrations obtained from overnight spot urine samples [23], but did not include measurements of 24-hour urinary

sodium excretion levels. While 24-hour urinary sodium excretion is widely recognized as the most accurate method for assessing dietary sodium intake [26,27], collecting 24-hour urine samples in large-scale population studies poses practical challenges. This method is prone to inaccuracies, often resulting from incomplete collection by participants. In the present study, we estimated calculated 24-h urinary sodium excretion levels using the following Tanaka's formulas (Eqs 1 and 2) [28].

**Eq 1) 24-h urinary creatinine excretion (mg)**

$$= -2.04 \times \text{Age (year)} + 14.89 \times weight \ (kg) + 16.14 \times height \ (cm) - 2244.45.$$

**Eq 2) Calculated 24-h urinary sodium excretion (mg)**

$$= 21.98 \times XNA^{0.392}.$$

$$XNA = 12 \times 24 - h \text{ urinary creatinine excretion (mg)}$$

This study categorized continuous variables, including serum cadmium, serum mercury, urinary arsenic, urinary arsenic-creatinine ratio, and urinary 24-hour sodium excretion levels, into low and high groups based on their median values. Furthermore, each variable was further divided into tertiles (1T, 2T, and 3T) to conduct additional analyses.

## Covariate

Information on sociodemographic factors such as age, sex, household income, education level, marital status, dietary energy, and potassium intake, as well as health-related factors including smoking habits, physical activity, hypertension, and diabetes mellitus, was gathered through medical examinations and self-reported questionnaires. Household income levels were divided into quartiles, ranging from quartile 1 (low) to quartile 4 (high). Education status was categorized as below middle school, high school, college, and higher than university. Marital status was defined as married or single. Drinking habit was classified as yes or no. Participants were also categorized by their smoking status as non-smokers or smokers. Moderate physical activity was recorded as yes or no. Hypertension was defined as the presence of high blood pressure or the use of antihypertensive medication, categorized as yes or no. Diabetes mellitus was categorized as yes or no, based on a history of the condition or diagnostic criteria (e.g., fasting glucose $\geq$ 126 mg/dL). Dietary energy intake was recorded in kilocalories per day (kcal/day), and dietary potassium intake was recorded in milligrams per day (mg/day).

## Statistical analysis

This study adhered to the KNHANES data analysis guidelines provided by the KCDC. KNHANES utilizes a complex sampling design incorporating stratification, cluster sampling, and sampling weights. Accordingly, PROC SURVEYLOGISTIC was employed to conduct analyses accounting for this complex sampling design.

We performed the Pearson chi-square test for categorical variables and the analysis of variance (ANOVA) for continuous variables. For this study, the dependent variable was defined as a categorical variable with three levels: underweight, normal weight, overweight, and obesity, based on BMI thresholds (underweight: BMI < 18.5 kg/m$^2$; normal: $18.5 \leq$ BMI < 23 kg/m$^2$; overweight and obesity: BMI $\geq$ 23 kg/m$^2$).

We utilized a polytomous logistic regression model to analyze the relationship between heavy metals, urinary sodium excretion, and overweight and obesity. Regression analyses incorporating continuous variables were also conducted. Additionally, restricted cubic splines were employed to visualize potential non-linear associations. This study was adjusted for the

clinical importance confounders, considered in previous study and study participants demographic: age, sex, household income, education level, marital status, smoking habits, moderate physical activity, dietary potassium (mg/day), dietary energy intake (mg/day), history of hypertension and diabetes mellitus.

Additionally, we performed an interaction analysis to evaluate whether the combined effects of urinary sodium excretion and serum mercury levels on overweight and obesity exceeded the sum of their individual effects. Multiplicative interaction was assessed by calculating the p-value for the interaction term in the logistic regression model. A p-interaction value of less than 0.05 was considered statistically significant, indicating a potential interaction between urinary sodium excretion and mercury levels in relation to the risk of overweight and obesity.

We examined multi-collinearity between independent variables with Pearson correlation coefficient and Variance inflation factor. The spline model was adjusted for age, sex, and confounding factors. Other statistical analyses were conducted using the SAS version 9.4. (SAS Institute Inc., Cary, NC, U.S.A) and a p-value < 0.05 was considered statistically significant.

## Results

The general characteristics of the participants are presented in Table 1. The participants included 6,609 adults with an average age of 45.0 years. Compared with underweight and normal participants, those with overweight and obesity were more likely to be older, male, physically inactive, smokers, married, and have higher urinary sodium excretion levels, serum mercury levels, dietary energy and potassium intake, and a greater prevalence of hypertension and diabetes mellitus.

The correlation matrix (S3 Table) showed statistically significant relationships between urinary sodium excretion, blood mercury levels, and BMI (p < 0.001). Additionally, the multicollinearity analysis using the Variance Inflation Factor (VIF) (S4 Table) indicated that all independent variables had VIF values below 2.0, confirming no significant multicollinearity.

Table 2 presents the associations between heavy metals, urinary sodium excretion levels, and the prevalence of overweight and obesity. The results indicate that participants with higher serum mercury levels ($\geq$ 5.1 µg/L) were significantly associated with increased odds of overweight and obesity (OR = 1.57, 95% CI = 1.31–1.88) compared to those with lower levels (< 3.0 µg/L). Similarly, participants with high urinary sodium excretion levels ($\geq$ 3588.8 mg/day) had significantly higher odds of overweight and obesity (OR = 2.21, 95% CI = 1.84–2.66) compared to those with low levels (< 2885.6 mg/day). These associations were consistent with the results of the linear regression analysis (S5 Table). Fig 2 illustrates the associations between heavy metals, urinary sodium excretion levels, and the odds of overweight and obesity. Serum mercury levels exhibit a potential non-linear trend, while urinary sodium excretion demonstrates a significant linear association, with higher levels correlating with increased odds of overweight and obesity. On the other hand, no statistically significant associations were observed for other heavy metals, such as serum cadmium and urinary arsenic, in relation to overweight and obesity.

The Table 3 presents the OR with 95% CI for overweight and obesity based on the combination of serum mercury levels and urinary sodium excretion levels. Participants with high urinary sodium and high serum mercury levels (3T/3T) had significantly higher odds of overweight and obesity compared to the reference group with low urinary sodium and low serum mercury levels (1T/1T) (OR = 3.61, 95% CI = 2.61–5.00, p-interaction < 0.01). Additionally, in the underweight group, participants with high urinary sodium and high serum mercury levels (3T/3T) had significantly lower odds of being underweight compared to the reference group with low urinary sodium and low serum mercury levels (1T/1T) (OR = 0.23, 95%

**Table 1. General characteristics of study subject, Korea National Health Examination and Nutritional Survey (KNHANES), 2008–2012.**

| | Normal (N = 4,225) | Underweight (N = 291) | Overweight & Obesity (N = 2,093) | P-value |
|---|---|---|---|---|
| | **N (%)** | **N (%)** | **N (%)** | |
| Sex | | | | |
| Male | 2053 (61.2) | 93 (2.8) | 1206 (36.0) | <0.01 |
| Female | 2172 (66.7) | 198 (6.1) | 887 (27.2) | |
| Monthly household income (KRW) | | | | |
| 1Q (< 1,500,000) | 1031 (63.2) | 67 (4.1) | 533 (32.7) | 0.04 |
| 2Q (1,500,000–2,999,000) | 1023 (61.9) | 64 (3.9) | 565 (34.2) | |
| 3Q (3,000,000–3,999,000) | 1056 (65.5) | 83 (5.1) | 473 (29.3) | |
| 4Q (≥ 4,000,000) | 1071 (65.2) | 75 (4.6) | 496 (30.2) | |
| Education status | | | | |
| Below Middle school | 1814 (61.7) | 71 (2.4) | 1053 (35.8) | <0.01 |
| High school | 973 (68.5) | 65 (4.6) | 383 (26.9) | |
| College | 840 (64.0) | 85 (6.5) | 387 (29.5) | |
| Higher than University | 581 (63.4) | 69 (7.5) | 266 (29.0) | |
| Marital status | | | | |
| Married | 3282 (62.8) | 164 (3.1) | 1780 (34.1) | <0.01 |
| Single | 936 (68.2) | 127 (9.2) | 310 (22.6) | |
| Drinking habit | | | | |
| No | 458 (62.6) | 28 (3.8) | 246 (33.6) | 0.42 |
| Yes | 3742 (64.1) | 258 (4.4) | 1837 (31.5) | |
| Smoking habits | | | | |
| No | 2294 (65.7) | 188 (5.4) | 1011 (28.9) | <0.01 |
| Yes | 1900 (61.9) | 98 (3.2) | 1072 (34.9) | |
| Moderate physical activity | | | | |
| No | 2712 (65.0) | 209 (5.0) | 1251 (30.0) | <0.01 |
| Yes | 1490 (62.1) | 78 (3.3) | 832 (34.7) | |
| Hypertension | | | | |
| No | 3279 (68.6) | 270 (5.6) | 1231 (25.7) | <0.01 |
| Yes | 944 (51.7) | 21 (1.2) | 860 (47.1) | |
| Diabetes mellitus | | | | |
| No | 3244 (68.2) | 256 (5.4) | 1259 (26.5) | <0.01 |
| Yes | 855 (51.6) | 24 (1.5) | 779 (47.0) | |
| | **Median (IQR)** | **Median (IQR)** | **Median (IQR)** | |
| Age (years) | 44 (24) | 31 (16) | 47 (22) | <0.01 |
| Urinary sodium excretion levels (mg/day) [a] | 3170.4 (1044.3) | 2692.9 (978.8) | 3445.6 (1148.9) | <0.01 |
| Urinary arsenic excretion levels (mcg/L) | 111.7 (122.5) | 101.6 (91.4) | 113.2 (120.1) | 0.14 |
| Urinary arsenic-creatinine ratio (µg /mg) | 0.8 (1.5) | 0.6 (1.3) | 0.8 (1.4) | 0.25 |
| Serum cadmium levels (µg/L) | 1.0 (0.7) | 0.8 (0.7) | 1.1 (0.7) | <0.01 |
| Serum mercury levels (µg/L) | 3.8 (2.9) | 3.1 (2.0) | 4.5 (3.7) | <0.01 |
| Dietary energy intake (mg/day) | 2005.5 (1442.8) | 1828.3 (1263.8) | 2085.2 (1508.1) | <0.01 |
| Dietary potassium intake (mg/day) | 3107.1 (2556.5) | 2602.4 (2154.0) | 3259.3 (2762.9) | <0.01 |

a. Urinary 24-hour sodium excretion levels were estimated by Tanaka equation in a spot urine.

CI = 0.06–0.88). Moreover, these results were consistent in the sensitivity analyses in the restricted study population (n = 6,137) excluding participants who exceeded the criteria for heavy metal safety limits (S1 and S2 Tables).

**Table 2. Odds Ratios (OR) and 95% Confidence Intervals (CI) for overweight and obesity based on levels of heavy metals and 24-hour sodium excretion in the healthy general population (N = 6,609)[a].**

| | Normal (N = 4,225) | Underweight (N = 291) | Underweight | Overweight & Obesity (N = 2,093) | Overweight & Obesity |
|---|---|---|---|---|---|
| | N (%) | N (%) | OR (95% CI)[b] | N (%) | OR (95% CI)[b] |
| **Serum cadmium levels (µg/L)** | | | | | |
| Low ($<$ 1.0) | 2127 (50.3) | 190 (65.3) | Ref | 986 (47.1) | Ref |
| High ($\geq$ 1.0) | 2098 (49.7) | 101 (34.7) | 0.80 (0.56–1.14) | 1107 (52.9) | 1.06 (0.91–1.24) |
| 1T ($<$ 0.7) | 1415 (33.5) | 143 (49.1) | Ref | 640 (30.6) | Ref |
| 2T (0.7–1.3) | 1241 (33.6) | 80 (27.5) | 0.75 (0.52–1.08) | 709 (33.9) | 1.07 (0.90–1.28) |
| 3T ($\geq$ 1.4) | 1389 (32.9) | 68 (23.4) | 0.82 (0.61–1.30) | 744 (35.5) | 1.05 (0.87–1.25) |
| **Serum mercury levels (µg/L)** | | | | | |
| Low ($<$ 3.9) | 2252 (53.3) | 199 (68.4) | Ref | 852 (40.7) | Ref |
| High ($\geq$ 3.9) | 1973 (46.7) | 92 (31.6) | 0.97 (0.68–1.39) | 1241 (59.3) | **1.51 (1.31–1.75)** |
| 1T ($<$ 3.0) | 1492 (35.3) | 141 (48.5) | Ref | 564 (26.9) | Ref |
| 2T (3.0–5.0) | 1448 (34.3) | 103 (35.4) | 1.10 (0.80–1.52) | 660 (31.5) | 1.10 (0.92–1.32) |
| 3T ($\geq$ 5.1) | 1285 (30.4) | 47 (16.2) | 0.81 (0.50–1.29) | 869 (41.5) | **1.57 (1.31–1.88)** |
| **Urinary arsenic excretion levels (mcg/L)** | | | | | |
| Low ($<$ 111.4) | 962 (49.9) | 83 (56.5) | Ref | 448 (49.1) | Ref |
| High ($\geq$ 111.4) | 966 (50.1) | 64 (43.5) | 1.20 (0.74–1.94) | 464 (50.9) | 0.99 (0.82–1.20) |
| 1T ($<$ 80.1) | 632 (32.8) | 54 (36.7) | Ref | 308 (33.8) | Ref |
| 2T (80.1–154.3) | 636 (33.0) | 56 (38.1) | 1.38 (0.81–2.36) | 306 (33.6) | 0.92 (0.72–1.19) |
| 3T ($\geq$ 154.3) | 660 (34.2) | 37 (25.2) | 1.14 (0.61–2.10) | 298 (32.7) | 0.89 (0.69–1.14) |
| **Urinary arsenic-creatinine ratio (µg /mg)** | | | | | |
| Low ($<$ 0.8) | 959 (49.7) | 82 (55.8) | Ref | 453 (49.7) | Ref |
| High ($\geq$ 0.8) | 969 (50.3) | 65 (44.2) | 1.04 (0.65–1.65) | 459 (50.3) | 1.00 (0.79–1.22) |
| 1T ($<$ 0.5) | 621 (32.2) | 66 (44.9) | Ref | 307 (33.7) | Ref |
| 2T (0.5–1.3) | 649 (33.7) | 37 (25.2) | 0.86 (0.51–1.46) | 312 (34.2) | 0.98 (0.77–1.26) |
| 3T ($\geq$ 1.3) | 658 (34.1) | 44 (29.9) | 1.03 (0.58–1.83) | 293 (32.1) | 0.86 (0.67–1.11) |
| **Urinary 24-hour sodium excretion levels (mg/day)[c]** | | | | | |
| Low ($<$ 3233.6) | 2247 (53.2) | 281 (74.9) | Ref | 839 (40.1) | Ref |
| High ($\geq$ 3233.6) | 1978 (46.8) | 73 (25.1) | **0.57 (0.40–0.82)** | 1254 (59.9) | **1.76 (1.51–2.05)** |
| 1T ($<$ 2885.6) | 1505 (35.6) | 182 (62.5) | Ref | 513 (24.5) | Ref |
| 2T (2885.6–3588.7) | 1453 (34.4) | 70 (24.1) | **0.49 (0.34–0.70)** | 685 (32.7) | **1.34 (1.11–1.61)** |
| 3T ($\geq$ 3588.8) | 1267 (30.0) | 39 (13.4) | **0.42 (0.27–0.67)** | 895 (42.8) | **2.21 (1.84–2.66)** |

Abbreviation: T, Tertile; OR, Odds ratio.

a. Analyses are adjusted for stratification, clustering, and sampling weights. Normal weight serves as the reference category.

b. Adjusted for age, sex, household income, education level, marital status, smoking habits, moderate physical activity, dietary potassium, dietary energy intake, history of diabetes and hypertension.

c. Urinary 24-hour sodium excretion levels were estimated by Tanaka equation in a spot urine.

## Discussions

The present study explored the association between heavy metals, urinary sodium excretion levels and obesity in a Korean population. This study suggests that participants with high serum mercury and high urinary sodium excretion levels are more likely to have overweight and obesity.

Regarding blood heavy metals, our results showed a suggestive association between higher cadmium levels and an increased prevalence of overweight and obesity, although this association did not reach statistical significance. A prospective cohort study in China reported no

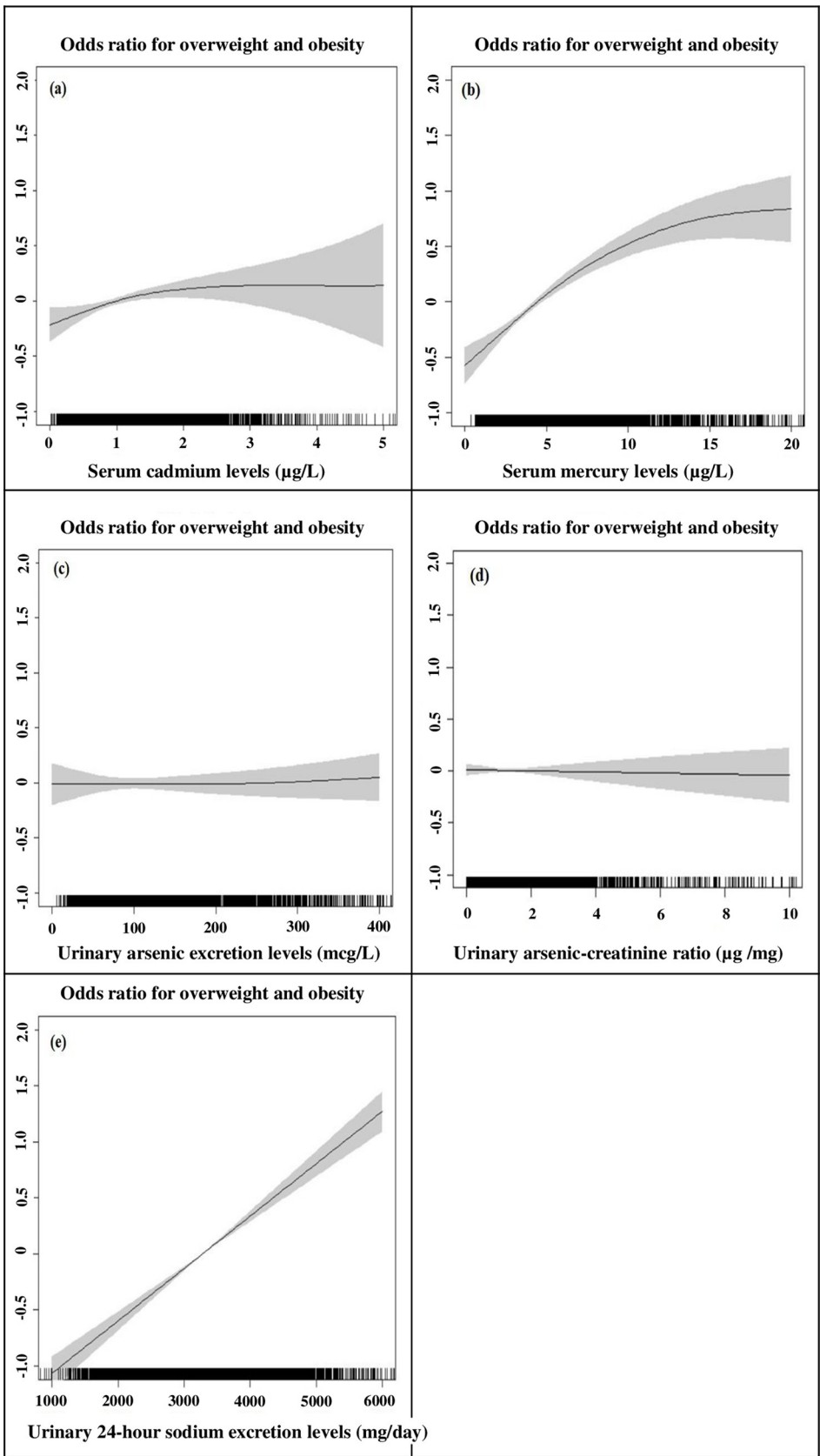

**Fig 2.** Odds ratios for overweight and obesity in relation to (a) serum cadmium levels (μg/L), (b) serum mercury levels (μg/L), (c) urinary arsenic excretion levels (mcg/L), (d) urinary arsenic-creatinine ratio (μg/mg), and (e) urinary sodium excretion (mg/day), modeled using restricted cubic splines.

association between cadmium exposure and obesity [16]. Another previous cross-sectional study conducted in a Northern Italy community demonstrated associations between serum cadmium and BMI values [29]. These findings are consistent with some previous studies, indicating a possible role of cadmium in adiposity. However, the lack of statistical significance in our study results may be attributed to the relatively small sample size in the obesity group, warranting caution in interpreting these findings.

In a case-control study carried out in Chinese adults, a positive correlation was identified between blood mercury levels and waist circumference (WC) [30]. Another cross-sectional study in Korean healthy subjects showed that the blood mercury concentration was significantly associated with visceral adipose tissue (VAT) [31]. In the present study, we observed a significant association between serum mercury levels and obesity, which is consistent with a previous study suggesting a potential obesogenic effect of mercury exposure. Notably, mercury, especially in its organic form, has been implicated in disrupting lipid metabolism and promoting adipo-genesis, possibly contributing to the observed association with obesity [32,33]. Further studies are warranted to confirm the present results and suggested mechanism.

Our study found that no significant difference in the odds of overweight and obesity between participants with high urinary arsenic excretion levels. Similarly, a cross-sectional study from 30 study sites in the United States, Ghana, South Africa, Seychelles, and Jamaica

**Table 3.** OR (95% CI) of overweight and obesity by the combination of serum mercury and urinary 24-hour sodium excretion in the general population (N = 6,609) [a].

| Urinary sodium levels [c] | Serum mercury levels | Normal (N = 4,225) N (%) | Underweight (N = 291) N (%) | Underweight OR (95%CI) [b] | Overweight & Obesity (N = 2,093) N (%) | Overweight & Obesity OR(95%CI) [b] |
|---|---|---|---|---|---|---|
| 1T | 1T | 591 (14.0) | 93 (32.0) | Ref | 143 (6.8) | Ref |
| 2T | 1T | 496 (11.7) | 59 (20.3) | 1.01 (0.64–1.61) | 164 (7.8) | 1.22 (0.85–1.75) |
| 1T | 2T | 503 (11.9) | 32 (11.0) | 0.92 (0.51–1.68) | 206 (9.8) | **1.64 (1.16–2.32)** |
| 2T | 2T | 490 (11.6) | 26 (8.9) | **0.49 (0.29–0.81)** | 202 (9.6) | **1.56 (1.10–2.21)** |
| 3T | 1T | 418 (9.9) | 30 (10.3) | **0.54 (0.29–0.99)** | 206 (9.8) | *1.40 (0.98–2.01)* |
| 1T | 3T | 399 (9.4) | 16 (5.5) | **0.38 (0.17–0.84)** | 277 (13.2) | **2.17 (1.53–3.08)** |
| 3T | 2T | 461 (10.9) | 12 (4.1) | **0.32 (0.16–0.68)** | 219 (10.5) | **2.27 (1.59–3.23)** |
| 2T | 3T | 462 (10.9) | 18 (6.2) | 0.64 (0.45–1.21) | 290 (13.9) | **2.58 (1.86–3.59)** |
| 3T | 3T | 406 (9.6) | 5 (1.7) | **0.23 (0.06–0.88)** | 386 (18.4) | **3.61 (2.61–5.00)** [d] |
| Low | Low | 1251 (29.6) | 151 (51.9) | Ref | 342 (16.3) | Ref |
| | High | 996 (23.6) | 67 (23.0) | 1.07 (0.69–1.65) | 497 (23.7) | **1.67 (1.35–2.08)** |
| High | Low | 1001 (23.7) | 48 (16.5) | **0.63 (0.41–0.97)** | 510 (24.4) | **1.93 (1.54–2.43)** |
| | High | 977 (23.1) | 25 (8.6) | **0.51 (0.28–0.93)** | 744 (35.6) | **2.69 (2.17–3.34)** |

Abbreviation: T, Tertile; OR, Odds ratio.

a. Analyses are adjusted for stratification, clustering, and sampling weights. Normal weight serves as the reference category.

b. Adjusted for age, sex, household income, education level, marital status, smoking habits, moderate physical activity, dietary potassium, dietary energy intake, history of diabetes and hypertension.

c. Urinary 24-hour sodium excretion levels were estimated by Tanaka equation in a spot urine.

d. P-interaction: p < 0.01.

observed that high arsenic in blood was not associated with increased odds of obesity after controlling for covariates [34]. In other studies, urinary marker of arsenic exposure was not associated with BMI among in rural India, or children in Mexico [35,36]. However, there is some scientific research that suggests a potential link between urinary arsenic levels and obesity. Arsenic is a naturally occurring element that can be found in water, soil, and food. It is known to be toxic and can have various negative health effects on humans [37]. In summary, while organic forms of arsenic in seafood are generally considered safe, excessive consumption of seafood could lead to increased exposure to both organic and inorganic arsenic, including more toxic forms. This is particularly relevant for populations with high seafood consumption, and there is a need for monitoring the potential health effects of long-term exposure to arsenic through diet [38].

In addition, we observed a significant positive association between higher urinary sodium excretion levels and the prevalence of overweight and obesity. Participants with elevated urinary sodium excretion had a higher risk of being overweight and obesity. This finding is consistent with previous research linking high sodium intake to weight gain and adiposity [39,40]. Excessive sodium intake may lead to fluid retention and increased thirst, potentially promoting overeating and caloric consumption, thus contributing to obesity development [41].

Considering the combined effects of blood heavy metals and urinary sodium levels, our study demonstrated that participants with high urinary sodium excretion and high serum mercury levels had the highest odds of overweight and obesity. Few mechanisms have been proposed to explain the potential links between these factors and the development of obesity. Heavy metals may potentiate the adverse effects of high sodium intake on adipocyte function, inflammation, and hormonal regulation, leading to an increased risk of obesity [42]. Also, high sodium intake has been shown to increase the gastrointestinal absorption of certain heavy metals, potentially leading to higher circulating levels of these toxicants [43]. Elevated blood levels of heavy metals, in turn may exacerbate their obesogenic effects. In summary, the biological basis linking blood heavy metals, urinary sodium, and obesity involves disruptions in adipocyte function, oxidative stress, inflammation, endocrine signaling, fluid balance, appetite regulation, and insulin resistance [42,44]. The interaction between heavy metal exposure and sodium intake may further amplify the risk of overweight and obesity. However, further studies are warranted to confirm the present results and suggested mechanisms.

While our study provides valuable insights, several limitations should be acknowledged. First, the cross-sectional design restricts causal inferences, and the potential for reverse causality cannot be ruled out. To minimize possibilities of reverse causality, we excluded participants with stomach cancer, liver cancer, colorectal cancer, lung cancer, stroke, and cardiovascular disease that potentially resulted in uremic malfunction and altered dietary habits after diagnosis. Second, in our study, we employed simple equations to calculate 24-hour urinary sodium excretion from spot urine samples. Since it is often impractical to directly measure 24-hour urinary sodium excretion, we relied on Tanaka's equations to estimate this value. Tanaka's equation is widely used in clinical practice in Korea [45,46]. Third, our study identified a potential interaction between urinary sodium excretion and blood mercury levels in relation to the risk of overweight and obesity. However, we were unable to clearly demonstrate a synergistic effect, and further quantitative analyses, such as the calculation of a synergy index, are required to confirm these findings [47]. Further research is needed to confirm and elucidate these interactions. Despite these limitations, the present study has important strengths. This study utilized representative survey data from the KNHANES, which is known for its robust study design. The KNHANES follows standardized protocols for data collection, ensuring the high quality and reliability of the data obtained. Second, our analysis incorporated a wide range of demographic, lifestyle, and health-related data as covariates in the statistical models.

These factors were carefully considered to control for potential confounding variables and ensure the robustness of our findings.

## Conclusions

In summary, higher mercury and higher urinary sodium excretion levels were associated with an increased likelihood of overweight and obesity. These findings suggest that heavy metal exposure and high sodium intake may be potential risk factors for overweight and obesity. Longitudinal studies and further research are warranted to validate and better understand the observed associations. Nevertheless, our study identified the association between the combined effects of sodium and heavy metals and obesity, which may contribute to the development of obesity prevention strategies for populations exposed to high-sodium diets and environmental pollution.

## Supporting information

**S1 Table. OR (95% CI) of overweight and obesity by the levels of heavy metals and 24-hour sodium excretion b in the healthy general population (N = 6,137).**
(DOCX)

**S2 Table. OR (95% CI) a of overweight and obesity by the combination of serum mercury and urinary 24-hour sodium excretion b in the general population (N = 6,137).**
(DOCX)

**S3 Table. Correlation matrix of independent variables in the study of heavy metals, urinary sodium excretion, and BMI levels.**
(DOCX)

**S4 Table. Evaluation of multicollinearity through Variance Inflation Factor (VIF).**
(DOCX)

**S5 Table. Linear regression analysis of heavy metals and urinary sodium excretion levels with BMI (continuous variables).**
(DOCX)

**S6 Table. Logistic regression analysis of heavy metal exposure and urinary sodium levels in relation to BMI including estimates, standard errors, and p-values.**
(DOCX)

## Author Contributions

**Conceptualization:** Sue K. Park.

**Data curation:** Sue K. Park.

**Formal analysis:** Juyeon Lee.

**Investigation:** Juyeon Lee.

**Methodology:** Juyeon Lee.

**Project administration:** Sue K. Park.

**Software:** Juyeon Lee.

**Supervision:** Sue K. Park.

**Validation:** Sue K. Park.

**Writing – original draft:** Juyeon Lee.

**Writing – review & editing:** Juyeon Lee, Sue K. Park.

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
