## [Decision Letter · Decision Letter 0]

7 Nov 2024

PONE-D-24-39306Associations of heavy metals and urinary sodium excretion with obesity in Adults: A cross-sectional study from Korean Health Examination and Nutritional SurveyPLOS ONE

Dear Dr. Park,

Thank you for submitting your manuscript to PLOS ONE. After careful consideration, we feel that it has merit but does not fully meet PLOS ONE’s publication criteria as it currently stands. Therefore, we invite you to submit a revised version of the manuscript that addresses the points raised during the review process. I agree with the reviewers that the authors need to elaborate on the rationale of the study and the statistical analysis. 

We look forward to receiving your revised manuscript.

Kind regards,

Iman Al-Saleh

Academic Editor

PLOS ONE

2. In the online submission form you indicate that your data is not available for proprietary reasons and have provided a contact point for accessing this data. Please note that your current contact point is a co-author on this manuscript. According to our Data Policy, the contact point must not be an author on the manuscript and must be an institutional contact, ideally not an individual. Please revise your data statement to a non-author institutional point of contact, such as a data access or ethics committee, and send this to us via return email. Please also include contact information for the third party organization, and please include the full citation of where the data can be found.

Reviewers' comments:

Reviewer's Responses to Questions

**Comments to the Author**

1. Is the manuscript technically sound, and do the data support the conclusions?

Reviewer #1: Yes

Reviewer #2: Partly

2. Has the statistical analysis been performed appropriately and rigorously? 

Reviewer #1: Yes

Reviewer #2: No

3. Have the authors made all data underlying the findings in their manuscript fully available?

Reviewer #1: Yes

Reviewer #2: Yes

4. Is the manuscript presented in an intelligible fashion and written in standard English?

Reviewer #1: Yes

Reviewer #2: Yes

5. Review Comments to the Author

**Reviewer #1**: This paper offers significant value by analyzing the association between sodium intake, heavy metals, and obesity using national data from the Korean Health Examination and Nutrition Survey (KNHANES). The study draws attention to the often-overlooked factors of sodium intake and heavy metal exposure in the context of obesity.

Major comment>

However, similar analyses have been conducted previously. For instance, the authors themselves have already reported on the association between urinary sodium levels and overweight/central obesity using the same KNHANES data in 2018 (Lee, J., Hwang, Y., Kim, KN. et al. BMC Nutr, 2018*). Additionally, other researchers have examined the association between mercury (Hg) and obesity using KNHANES data (Eom, SY., Choi, SH., Ahn, SJ. et al. Int Arch Occup Environ Health, 2014*). There is also a meta-analysis on the link between Hg and obesity (Jeon J, Park K. Korean Journal of Community Nutrition, 2023). Therefore, the introduction of this paper needs further elaboration on the rationale for analyzing sodium and heavy metals together. Why is it necessary to study the combined effects of sodium and heavy metals in relation to obesity?

In addition, please clarify whether the findings support a synergistic effect between urinary sodium excretion and heavy metal exposure on obesity, or whether the relationship is simply additive. Can the results of this study distinguish between these two effects? Please explain the interaction analysis (p-interaction) shown in Table 3 in the Methods section. Additionally, the results of “We examined multi-collinearity between independent variables with Pearson correlation coefficient and Variance inflation factor.” are not presented in the manuscript.

Minor comment>

1. Study Population

- Clarify whether the study population included only adults or also included children and adolescents. Table 1 mentions the exclusion of participants with chronic kidney failure and chronic liver disease, which should also be explicitly stated in the Materials and Methods section. Given the use of urinary sodium and arsenic analysis, excluding participants with chronic kidney disease is appropriate.

3. Method

- Since this study primarily used calculated 24-hour urinary sodium excretion levels, please provide the formula used in the Materials and Methods section.

4. Table 1

- If dietary potassium and dietary energy intake were adjusted for in the logistic regression models, it would be helpful to include these variables in Table 1.

- For continuous variables like sodium, arsenic (As), cadmium (Cd), and mercury (Hg), presenting the data as median (IQR) might be more appropriate.

- There is a discrepancy between the p-value for arsenic in Table 1 and the results discussed in the main text. Please correct this inconsistency.

5. Table 2

- The table appears overly complex. If you have performed logistic regression for underweight, overweight, and obesity groups, the "normal" column may not be necessary. Instead, consider focusing on logistic regression analysis for "obesity" or a combined "overweight+obesity" group. Additionally, the "Total" column seems redundant and could be removed. If you categorized the variables (low/high) based on the median values, please describe this in the Methods section or Table footnote.

- In Table 1, variables such as smoking status, marital status, income, and education showed significant differences across BMI categories. It is unclear why these variables were not adjusted for in the regression models. Please clarify this choice.

6. Figure 1

- Figure 1 is not referenced in the main text, and there is a mismatch between the numbers in Figure 1 and those provided in the Methods section. Please ensure consistency between the figures and the text.

**Reviewer #2**: This is a really interesting and important manuscript. The authors tried there best to describe the entire process so that reader can get the take home message and can replicate similar studies. But to consider it for publishing, a major revision is required. The following are the overall areas where the authors should focus and submit the revised version.

1. The research objective was not clearly stated. Is this study to identify novel risk factors? or just to explore the association between heavy metal, urinary sodium excretion and obesity and overweight. The author should clearly describe it.

2. The author mentioned that the aim was to identify novel risk factors (line 82-83)... but in the result the author didn't highlight what was/were the novel risk factor and why that is novel?

3. In the statistical analysis section, the author mentioned that, they used polytomous logistic regression model but in the result section reported results on spline. These needs to be clearly explained. Moreover, there is a lack of description of the outcome (dependent) variable for the logistic regression model. The statistical analysis part should be revisited with clearly defined the dependent and covariates for the model and perform the analysis accordingly. Also, it is highly recommended to address clustering effect due to sampling techniques. A robust standard error should be computed and all the inference should be upon robust standard errors.

4. I would also recommend to do a linear regression without categorising the body weight variable and add that as a supplementary information.

5. The tables should be revisited, the message from the table is not easy to understand by readers. It should be revisited.

As a minor comments, the author should add more recent work related to urinary sodium excretion and obesity. There are recent publication out there that are more recent that the author cited in this manuscript.

Line 65: Recent investigations have suggested a potential link between heavy metal exposure and obesity. This statement should supported by results and a proper citation.

6. PLOS authors have the option to publish the peer review history of their article (what does this mean?). If published, this will include your full peer review and any attached files.

Reviewer #1: No

Reviewer #2: No

---

## [Author Response · Author response to Decision Letter 0]

9 Dec 2024

We sincerely appreciate the reviewers' valuable comments. Please refer to the attached file, "Response to Reviewers," for our detailed responses. Thank you for your thoughtful consideration.

---

## [Decision Letter · Decision Letter 1]

20 Dec 2024

PONE-D-24-39306R1Associations of heavy metals and urinary sodium excretion with obesity in Adults: A cross-sectional study from Korean Health Examination and Nutritional SurveyPLOS ONE

Dear Dr. Park,

Thank you for submitting your manuscript to PLOS ONE. Reviewer #1 has provided some comments that need clarification before a final decision can be made. You will find the reviewer's comments at the end of this email.  Therefore, we invite you to submit a revised version of the manuscript that addresses the points raised during the review process.

We look forward to receiving your revised manuscript.

Kind regards,

Iman Al-Saleh

Academic Editor

PLOS ONE

Journal Requirements:

Reviewers' comments:

Reviewer's Responses to Questions

**Comments to the Author**

1. If the authors have adequately addressed your comments raised in a previous round of review and you feel that this manuscript is now acceptable for publication, you may indicate that here to bypass the “Comments to the Author” section, enter your conflict of interest statement in the “Confidential to Editor” section, and submit your "Accept" recommendation.

Reviewer #1: All comments have been addressed

2. Is the manuscript technically sound, and do the data support the conclusions?

Reviewer #1: Yes

3. Has the statistical analysis been performed appropriately and rigorously? 

Reviewer #1: Yes

4. Have the authors made all data underlying the findings in their manuscript fully available?

Reviewer #1: Yes

5. Is the manuscript presented in an intelligible fashion and written in standard English?

Reviewer #1: Yes

6. Review Comments to the Author

**Reviewer #1**: The authors have adequately addressed the comments raised in the previous round of review. I believe the manuscript has improved significantly.

There is still some uncertainty about whether sodium and mercury really have a synergistic effect. In Table 3, the odds ratio (OR) for the combination of sodium 3T/mercury 3T is 3.61, which is the highest. However, the OR for Sodium 3T/Mercury 1T is 1.40 and for Sodium 1T/Mercury 3T it is 2.17. The OR of 3.61 is not significantly greater than the sum of the ORs of 1.40 and 2.17. Although the reported p for interaction suggests an interaction between the two variables, this does not necessarily confirm a synergistic effect. However, the authors have appropriately acknowledged in the DISCUSSION section that further research is needed to confirm whether this represents a synergistic effect. This limitation has been well addressed and is acceptable.

I have a few minor suggestions for further refinement:

Line 228:

In Figure 23, the relationship between urinary sodium excretion levels and overweight/obesity appears linear. Is it non-linear?

Line 236:

There is an error in reporting the OR and 95% CI. The correct value should be OR 3.61 (2.61–5.00) instead of OR 0.23 (0.06–0.88). Please revise this accordingly.

7. PLOS authors have the option to publish the peer review history of their article (what does this mean?). If published, this will include your full peer review and any attached files.

Reviewer #1: No

---

## [Author Response · Author response to Decision Letter 1]

20 Dec 2024

Thank you for your valuable comments. Please refer to the attached 'Response to Reviewer' file.

---

## [Editor Report · Decision Letter 2]

23 Dec 2024

Associations of heavy metals and urinary sodium excretion with obesity in Adults: A cross-sectional study from Korean Health Examination and Nutritional Survey

PONE-D-24-39306R2

Dear Dr. Park,

We’re pleased to inform you that your manuscript has been judged scientifically suitable for publication and will be formally accepted for publication once it meets all outstanding technical requirements.

Kind regards,

Iman Al-Saleh

Academic Editor

PLOS ONE
---

## [Editor Report · Acceptance letter]

16 Jan 2025

PONE-D-24-39306R2 

PLOS ONE

Dear Dr. Park, 

I'm pleased to inform you that your manuscript has been deemed suitable for publication in PLOS ONE. Congratulations! Your manuscript is now being handed over to our production team.

Kind regards, 

on behalf of

Dr. Iman Al-Saleh 

Academic Editor

PLOS ONE